# The probability of *Plasmodium vivax* acute illness following primary infection and relapse in Papua New Guinea

Amanda Ross[1,2]*, Cristian Koepfli[1,2¤], Lincoln Timinao[3], Diggory Hardy[1,2], Tobias Thüring[1,2], Melanie Loeffel[1,2], Benson Kiniboro[3], Ingrid Felger[1,2], Ivo Mueller[4,5]

**1** Swiss Tropical and Public Health Institute, Allschwil, Switzerland, **2** University of Basel, Basel, Switzerland, **3** Papua New Guinea Institute of Medical Research, Goroka, Papua New Guinea, **4** Department of Medical Biology, University of Melbourne, Parkville, Australia, **5** Walter and Eliza Hall Institute of Medical Research, Melbourne, Australia

¤ Current address: University of Notre Dame

\* amanda.ross@unibas.ch

## Abstract

Inoculation with *Plasmodium vivax* malaria parasites can lead to blood-stage infections from the primary infection and relapses from liver-stage parasites or non-circulating merozoites. Understanding the risk of clinical illness following primary infection and relapse would inform surveillance and intervention strategies, but the probabilities are uncertain in people living in endemic areas. A major difficulty lies in the inability to distinguish primary infections and relapses. In this study, we estimate the probabilities of clinical illness using the different seasonal patterns of primary infection and relapse. Children aged one to three years in Ilaita, Papua New Guinea, were followed up over 16 months for illness (fever with ≥500 parasites/µl) with fortnightly active and passive case detection, and for blood-stage infection every two months. Estimates of the number of primary infections and relapses for each two-month time-period, age-group, village and ITN use category were derived from previous analyses using genotyping data. In this study, we use a Bayesian statistical model to relate the number of observed *P. vivax* clinical cases in each covariate category to the expected numbers of primary infections and relapses. We include the cumulative number of primary infections experienced since birth as a proxy for acquired immunity. To reflect uncertainty, we use varying assumptions about whether relapses can cause illness in different circumstances. The probability of illness decayed exponentially with increasing cumulative numbers of primary infections experienced. The estimated probability of illness following relapses was lower than that for primary infection, how much lower depended on how they were defined. Later relapses within the same brood tended to have lower probabilities than earlier ones. Varying seasonally, relapses were estimated to contribute half of *P. vivax* illness in this cohort despite accounting for 80% of the force of blood-stage infection. The results can inform estimates of the burden of

**Data availability statement:** Data and code are available at: https://github.com/rossaman4/vivax_clinical_illness.

**Funding:** This study was supported by the Gottfried und Julia Bangerter-Rhyner Stiftung (to AR), the Novartis Foundation for Medical Biological Research project number 13A13 (to AR), the Swiss National Science Foundation project number 320030-125316, and the Gates Foundation project number OPP1032350, and National Health and Medical Research Council (NHMRC) Senior and Principal Research Fellowships (1043345 and 1155075, both to IM). The data used were additionally supported by the Swiss National Science Foundation project number 31003A-112196, the National Institutes of Health (AI063135, AI46919, and TW007872) and the Australian Agency for International Development. Infrastructure was supported by the Victorian State Government OIS and the NHMRC IRIIS grants. The funders had no role in study design, data collection and analysis, decision to publish, or preparation of the manuscript.

**Competing interests:** The authors have declared that no competing interests exist.

*P. vivax* and provide building blocks for mathematical models for predicting the impact of interventions. Interventions triggered by clinical cases would focus on more recent infections and age-groups with less acquired immunity.

## Author summary

The probability that a pathogen causes illness or is tolerated without symptoms is important for quantifying the burden of disease, designing surveillance and intervention strategies, and optimizing resource allocation. In the case of *Plasmodium vivax*, one of the malaria species to infect humans, blood-stage infections may be caused by the primary infection typically shortly after an infectious mosquito bite or by relapses weeks or months later from developmentally-arrested liver-stage parasites or potentially from non-circulating parasites in other organs.

To estimate the probability of illness following primary infection and relapse, we analysed data from children in Papua New Guinea who were followed up for 16 months. Our previous analyses of genotyping data from these children estimated the differing seasonal patterns of primary infections and relapses. We combine these with the recorded numbers of clinical cases in a statistical model to estimate the probability of illness following primary infection and relapse, using the number of cumulative primary infections as a proxy for acquired clinical immunity.

We estimated a decline in the probability of illness with increasing cumulative exposure for both primary infections and relapses. There were also lower estimated probabilities following relapses compared to primary infections. Varying seasonally, relapses were estimated to contribute half of *P. vivax* illness in this cohort despite accounting for 80% of the force of blood-stage infection. Interventions triggered by clinical cases would focus on more recently transmitted infections and age groups with less acquired immunity.

## Introduction

Despite major progress over the past 25 years, *Plasmodium vivax* malaria remains a substantial public health challenge with an estimated nine million clinical cases in 2023 [1]. An estimated 3.3 billion people were living within the limits of *P. vivax* transmission in 2017 [2] with costs of clinical cases estimated at US$350 million in that year [3].

The outcomes of a bite from a mosquito carrying infectious *P. vivax* parasites range from asymptomatic infection to uncomplicated clinical malaria, severe cases and death [4]. The majority of *P. vivax* blood-stage infections are asymptomatic and often submicroscopic [5,6]. The role of the low-density infections in sustaining residual transmission is unclear. Although most of them do carry gametocytes (the sexual stage transmissible to mosquitoes) [7], infectivity of *P. vivax*-positive blood

samples remains low at parasite densities below 10–100 parasites per microliter [8]. Due to the rapid initial development of gametocytes and strong association between asexual parasite and gametocyte densities, it is thought that symptomatic *P. vivax* infections contribute disproportionally to transmission [8–10].

*P. vivax* is thought to be more resilient than *P. falciparum* to conventional control measures, and this is often attributed to the ability to relapse from long-lasting developmentally arrested liver-stage forms (hypnozoites) [4,11]. Relapses have been assumed to pose a reduced risk of causing illness, given the rapid acquisition of immunity to *P. vivax* [12] and the genetic relatedness of primary and relapsing infections [13,14]. Historical data from patients deliberately infected with *P. vivax* to treat neurosyphilis showed a lower probability of acute illness following a relapse or reinfection with the same strain compared to a different strain [12,15–17]. However, relapses may still cause a substantial burden. Studies that treated some participants to remove hypnozoites reported a substantial reduction in clinical episodes [18,19].

The probability of illness following blood-stage infection is an important parameter for designing strategies to control and eliminate *P. vivax* malaria, parameterizing building blocks of mathematical models to predict the likely impact of interventions and surveillance, interpreting routine data from health facilities [20–22] and informing estimates of the burden of disease. Predictions of the impact of interventions triggered by clinical illness, such as case-management or reactive response, rely on accurate estimates.

The proportion of *P. vivax* blood-stage infections resulting in acute illness in endemic areas is not well established. Age-incidence curves in areas with moderate or high transmission show that younger children have a higher risk of clinical episodes than older children, suggesting that clinical immunity is acquired gradually through exposure [23]. Statistical relationships between measures of clinical illness and parasite densities [24] or prevalence [22,25] have been estimated. The proportion of individuals who were symptomatic of those with PCR-detected parasitaemia in community surveys in low transmission settings in the Amazon was estimated to be between 7% and 42% in different locations [26]. However, to gain the probability of illness per blood-stage infection would require estimates of the force of infection, sub-patent infections, treatment-seeking behavior and the durations of illness and blood-stage infection. A modelling study based on routine clinical data in a low transmission setting in the Amazon estimated that 5% to 34% of all blood-stage infections resulted in clinical malaria [27].

Estimating the probability of illness triggered by primary infection and relapse is challenging. Health centre records lack key information on infection rates and the probability of seeking treatment. Cohort studies in the community are generally only feasible in moderate to high transmission settings due to the need for sufficient numbers of clinical cases. While individual broods (we use brood to mean the primary infections and relapses stemming from related parasites in the same mosquito bite) can be distinguished using genotyping, distinguishing between primary infections and relapses from the same brood is not possible. Methods have been developed to assign individual recurrences as relapses probabilistically under certain circumstances: using the genotype and time since presenting for treatment in antimalarial drug trials, and in low transmission settings [28–30].

We previously estimated the seasonal force of blood-stage infection from primary infections and relapses on a population level in a cohort of children in a relatively high transmission setting in Papua New Guinea [31]. In the present study, we build on our previous work to estimate the probability of illness following primary infection and relapses in the same cohort. We relate the number of clinical cases detected via active and passive case detection in each age-group and time-period to the expected numbers of primary infections and relapses. We account for acquired immunity by using the expected number of previous primary infections experienced as a proxy.

## Methods

### Ethics statement

The cohort study was approved by institutional review boards of the Papua New Guinea Medical Research Advisory Committee (approvals 05.19 and 09.24), University Hospitals Case Medical Center (Cleveland, Ohio USA), and the Ethikkommission beider Basel (approval 03/06). Informed written consent was provided by the parents or legal guardians of each child.

## Data from the cohort in Ilaita, Papua New Guinea

The cohort was based in Ilaita, Maprik district, East Sepik Province, Papua New Guinea and has been described fully elsewhere [23,32–35]. Briefly, 264 children aged between one and three years at enrolment were followed up over 16 months. The prevalence by microscopy at enrolment was 44% for *P. vivax* and 33% for *P. falciparum*.

Illness was detected through active case detection with fortnightly visits to the child's home. Additionally, carers were encouraged to bring the child to the study clinic when they were ill. *P. vivax* acute illness was defined as a density of 500 parasites/μL or greater by microscopy with a fever of 37.5°C or reported fever within the last 48 hours. Overall there were an estimated 2.46 *P. vivax* and 2.56 *P. falciparum* episodes per child per year [23].

Artemether-lumefantrine was provided by the study team to those who were RDT or microscopy positive and had fever or history of fever. Further antimalarial drugs could be provided outside the study and were recorded in each child's health book. At the time of the study, radical cure with primaquine was not part of the national treatment guidelines and was not administered.

Finger prick blood samples were collected for genotyping at nine routine survey time-points at two-monthly intervals. The first and last surveys collected a single blood sample while at the second to eighth routine time-points, two samples were taken 24 hours apart (originally to assess the detectability of parasite clones [32]). Additionally, blood samples were taken if the child was ill, either when presenting for treatment or during the active case detection visits. Blood samples that were positive by microscopy or LDR-FMA, a molecular method for species detection, were genotyped for *P. vivax* using microsatellites *msp1*F3 and MS16 and for *P. falciparum* using the marker *msp2* via capillary electrophoresis [33,36].

This study includes data from 243 (92%) children who were present at six or more routine survey time-points.

## Classifying relapses

We use the term 'relapse' widely to include the release of parasites into the bloodstream from hypnozoites in the liver or from non-circulating blood-stage merozoites [37]. The majority of the *P. vivax* biomass is cryptic in the spleen and bone marrow rather than circulating [38].

Our force of infection contributed by relapses was estimated from genotyping data from the Ilaita cohort, accounting for imperfect detection. We assumed that relapse rates are unaffected by the presence of an ongoing blood-stage infection from the same brood [31](S1 Appendix).

To capture uncertainty, we developed eight classifications of relapses. These classifications vary according to three assumptions: whether the relapse could cause illness if a blood-stage infection from the same brood is present, whether a relapse could cause illness if the primary infection was asymptomatic, and whether the probability of illness varies by the order of the relapses since the primary infection. While the classifications vary, the subsequent modelling methods remain the same.

*Assumptions about whether a relapse could cause illness if a blood-stage infection by the same brood is present.* We use two definitions to represent the two extremes. For **relapse definition A**, all relapses are assumed to potentially cause illness. For **relapse definition B**, we assume that the relapses cannot cause illness if a blood-stage infection by the same brood is already present. Additionally for relapse definition B, if a relapse occurs a short time after a blood-stage infection from the same brood has cleared naturally without treatment, then we assume that the new blood-stage infection does not establish since specific immunity would remain. In the absence of data on the decay of *P. vivax* immunity, we arbitrarily assume that this period lasts for 20 days.

For both relapse definitions, relapses occurring during the prophylactic period of antimalarial drug treatment are not counted.

*Assumptions about whether a relapse may cause illness if the primary infection did not.* It is not known whether relapses may cause illness if the primary infection did not. If the male and female gametes came from the same parasite clone (selfing), then the hypnozoites have no genetic variation (apart from natural mutation). In the case of recombination, the primary

infection is thought to contain greater genetic variation than each of the subsequent relapses of the same brood [39]. We include both possibilities, allowing relapses to cause illness regardless of, or only if, the primary infection caused illness.

*Allowing the probabilities of illness to vary by whether the relapse came earlier or later after the primary infection.* We additionally divide into earlier and later relapses, to allow the probabilities of illness to vary by their order. For relapse definition A, we use two categories: the first and second relapses, and the third or later relapses. For relapse definition B, we group the relapses into the first relapse and then the second or later relapse. The categories were based on the numbers of relapses. We additionally use one category with early and late relapses taken together.

In total, there were eight different classifications (Table 1).

## Modelling strategy

We estimate the probabilities of *P. vivax* illness following primary infection and relapse taking acquired immunity into account. Our strategy consists of three components.

First, we obtain the expected numbers of primary infections and relapses for each combination of age-group, ITN use, village and two-month time interval. The differing seasonal patterns of primary infections and relapses in the Papua New Guinea cohort provide information to disentangle illness caused by primary infections and by relapses. The previous estimates of the forces of blood-stage infection from primary infections and relapses were input into a simulation model in order to gain the expected numbers of primary infections and relapses taking the different relapse classifications, antimalarial treatments and prophylactic periods into account.

In a second step, we relate the observed numbers of *P. vivax* illness in the covariate categories to the expected numbers of primary infections and relapses. We include the expected number of lifetime primary infections (or equivalently broods) as a proxy for acquired clinical immunity. This provides the estimated probabilities of illness.

Finally, we include the two parts as a module in OpenMalaria [40], a comprehensive simulator of malaria, allowing us to use previously developed components for mosquito biting [41,42]. We predict age-incidence curves for different transmission intensities and validate the model using observed age-curves from sites with the same relapse pattern.

The code and data are available at www.github.com/rossaman4/vivax_clinical_illness.

## The expected numbers of primary infections and relapses

Estimates of the force of infection from primary infections and relapses were obtained from previous analyses that used genotyping data from symptomatic and asymptomatic infections in the same cohort [31]. Briefly, the seasonal pattern of

**Table 1. Summary of relapse classifications.**

| Relapse classification | Relapse definition | Categories of order of relapses | Which relapses can cause illness |
|---|---|---|---|
| A1j | A | 1: One category only | j |
| A1k | A | 1: One category only | k |
| A2j | A | 2: Two categories: First and second, third or later relapses | j |
| A2k | A | 2: Two categories: First and second, third or later relapses | k |
| B1j | B | 1: One category only | j |
| B1k | B | 1: One category only | k |
| B2j | B | 2: Two categories: First, and second or later relapses relapses | j |
| B2k | B | 2: Two categories: First, and second or later relapses relapes | k |

A: all relapses.

B: relapses only in the absence of a blood-stage infection by the same brood and 20 days after natural clearnace.

j: all relevant relapses.

k: relapses only where the primary infection caused illness.

primary infections was assumed to follow the seasonal pattern of *P. falciparum* infections since the same mosquito vectors transmit both species. In order to estimate the force of infection of primary infection and relapses, additional sources of data were used. The timing of *P. vivax* relapses was estimated using from data from prison volunteers infected with the Chesson strain which originated from the same island of New Guinea [43], and the relative biting rates in children of different ages was calculated using body surface area derived from the WHO growth standards [44,45]. Together these sources provided estimates of the seasonal force of infection of *P. vivax* primary infections and relapses. The genotyping data is not used again, avoiding double-dipping.

To obtain the expected numbers of primary infections and relapses experienced on average in each covariate category (combination of age-group, ITN use, village and time-period), we needed to take into account treatment, prophylactic periods and the different definitions of relapses. To do this, we used an individual-based simulation model parameterized with the previous analyses (S1 Appendix).

The duration of blood-stage infection is not well known for *P. vivax*, and different estimates exist in the literature. To reflect uncertainty, we use two different estimates (S1 Appendix) and consequently have two different input parameter value sets.

**Statistical model to estimate the probability of acute illness from the expected numbers of relapses and primary infection**

We first describe model variant A1. There are four model variants (A1, A2, B1, B2) following the relapse classifications (Table 1), and two different sets of parameter values.

**Model variant A1: A single group of relapses using relapse definition A**

The observed number of *P. vivax* clinical cases, $c(a, t)$, in covariate category $a$ in time interval $t$ was assumed to follow a Poisson distribution:

$$c(a, t) \sim Poisson\left(n(a, t) \cdot p(a, t)\right)$$

where $p(a, t)$ is the overall probability of clinical illness, and $n(a, t)$ is the sum of the expected numbers of primary infections, $n_p(a, t)$, and relapses, $n_r(a, t)$, so that $n(a, t) = n_p(a, t) + n_r(a, t)$. We assume that the illness occurs at the beginning of the blood-stage infection, recognizing that this is a simplifying assumption.

The overall probability of clinical illness, $p(a, t)$, is a weighted sum of the contributions from primary infection and relapses:

$$p(a, t) = \frac{n_p(a, t)}{n(a, t)} \, p_p(a, t) + \frac{n_r(a, t)}{n(a, t)} \, p_r(a, t)$$

where $p_p(a, t)$ and $p_r(a, t)$ are the probabilities of clinical illness following primary infection and relapses. For all covariate categories in the cohort, $n(a, t) > 0$ and so $p(a, t)$ is always defined. If $n(a, t) = 0$, then there would be no children with *P. vivax* primary infections or relapses in that covariate category and the probability of illness would not be relevant.

The probabilities of illness are allowed to vary by the expected number of cumulative primary infections as a proxy for acquired clinical immunity. We explored different decays in preliminary analyses and found an exponential decay with cumulative infections to have the best fit, so that:

$$p_p(a, t) = p_{p0} \, e^{- X(a,t) \, X^*}$$

$$p_r(a, t) = p_{r0} \, e^{- X(a,t) \, X^*}$$

where $X(a, t)$ is the expected cumulative number of primary infections experienced by a child in covariate category $a$ at the mid-point of time interval $t$ assuming a repeating seasonal pattern. The constants $p_{p0}$, $p_{r0}$ are the probabilities of illness in naïve individuals in the absence of maternal immunity. $X^*$ is a constant.

Model variant A1k assumes that relapses can only cause illness if the primary infection of the same brood did. The probability of illness for relapses under this assumption is approximated by $\frac{p_r(a,t)}{p_p(a,t)}$. While it is possible to account for the difference between the cumulative number of primary infections at the time of the relapse and at the time of the primary infection of the brood, this added complexity did not substantively alter the results and was therefore omitted.

We constrain $p_{p0} \geq p_{r0}$. $X^*$ is constrained to have the same value for primary infections and relapses. The constraints ensure that primary infection has an equal or greater probability of triggering a clinical episode than the subsequent relapses from the same brood and ensure identifiability. The parameters are described in Table 2.

We fit the model in a Bayesian framework using the Hamiltonian Monte Carlo algorithm implemented in Stan [46] via the R [47] interface package RStan [48].

**Model variant B1.** Model variant B1 differs from model variant A1 only in the definition of relapses. Relapse definition B excludes relapses occurring when there is a blood-stage infection by the same brood already present or during an immune period after natural clearance.

**Model variants A2 and B2.** The model is extended to incorporate three categories (Table 1) in the equation for $p(a, t)$.

Table 2. **Quantities in the statistical model to estimate the probabilities of clinical illness (Model variant A1).**

| Quantity | Description | Source |
|---|---|---|
| **Indices** | | |
| $t$ | Two-month time interval of the cohort study | – |
| $a$ | Covariate category for combination of age-group, village, and ITN use[1] | – |
| **Parameters estimated in the Bayesian model** | | |
| *Probability of illness following blood-stage infection in naïve individuals in the absence of maternal immunity* | | |
| $p_{p0}$ | following primary infection | To be estimated |
| $p_{r0}$ | following relapse[2] | To be estimated |
| $X^*$ | Constant for exponential decay | To be estimated |
| **Quantities input to the Bayesian model** | | |
| *Number of clinical cases following blood-stage infections for a child in category a in interval t* | | |
| $c(a, t)$ | Number of *P. vivax* clinical cases | cohort data [23] |
| *Denominators* | | |
| $n_p(a, t)$ | expected number of primary infections for category $a$ at time interval $t$ | S1 Appendix |
| $n_r(a, t)$ | expected number of relapses | S1 Appendix |
| $n(a, t)$ | expected number of primary infections or relapses | $n(a, t) = n_p(a, t) + n_r(a, t)$ |
| $X(a, t)$ | expected cumulative number of primary infections in child in covariate category $a$ at time interval $t$ | S1 Appendix |

[1] Age-groups: 1-<1.5,1.5-<2,2-<2.5,2.5-<3,3-<3.5,3.5-<4,4 or more years. ITN use: use of ITN on <50% or $\geq$ 50% of nights, village: Sunuhu, Ilaita.

[2] Assuming that relapses can cause illness whether the primary infection did or not (j). For relapses only if the primary infection caused illness (k), the probability is represented by $\frac{p_{r0}}{p_{p0}}$.

## Predicting *P. vivax* clinical illness in all age-groups and transmission intensities

To predict age-incidence curves, we adapted the framework of *OpenMalaria* [40], a comprehensive simulator originally developed for *P. falciparum* malaria. We integrated a module for *P. vivax* into *OpenMalaria* (release 48), comprising our simulation model for primary infections and relapses (S1 Appendix) together with our estimates of the probability of clinical illness. We added components for the latent period (the time between the infective bite and the primary infection), maternal clinical immunity, and the relationship between transmission intensity and the force of primary infection. Using *OpenMalaria* provides us with existing components for infective mosquito bites and established machinery for generating predictions. Although sufficient for our purpose, this does not provide a full *P. vivax* transmission model. We use fixed transmission intensities as inputs to our specific simulations and do not require dynamical feedback effects.

The primary blood-stage infection occurs immediately following the latent period. For tropical *P. vivax* strains, the median incubation period has been estimated as 12 days (95% CI 12–12) and 95th centile of 16 days (15–17) using data from neurosyphilis patients who were deliberately infected as malaria therapy [49]. The incubation period was not associated with sporozoite dose for analyses involving the Chesson strain [50]. Similar estimates are reported from recent challenge studies with sporozoite-induced infections in malaria-naïve adults [51]. We set the simulated latent period to three five-day time-steps. (The latent period had not been included in our statistical model since primary infections rather than inoculations were used as the starting point).

We add a component for maternal immunity against clinical illness. We use a Hill function, the same functional form of a model for maternal immunity against *P. falciparum* asexual parasites in infants by age [52].

We assume that there is heterogeneity in the number of infectious bites [41,42]. For *P. vivax* there is little available data, and so we use both within- and between-host heterogeneity for plausibility [53].

## Validation

We validated the model for *P. vivax* clinical illness using observed age-incidence patterns from settings with the same relapse pattern according to Battle *et al* [54]. We used age-incidence curves from health facility data in the Wosera [24,55] and other sites in Papua New Guinea [56], Papua Indonesia and Vanuatu [57] and estimates of the incidence in older children in cohorts in Papua New Guinea [19,58,59]. The component for maternal immunity was validated using age-incidence curves from the placebo arm of a trial of intermittent preventive treatment in infants [60]. We acknowledge that age-dependent treatment-seeking behavior may influence the health facility patterns [61].

## Results

### The incidence of primary infections and relapse

Estimates of the incidence and seasonality of primary infection and relapse were derived from previous estimates of the forces of infection obtained using genotyping data, and adjusted using the simulation model to account for prophylactic periods, treatment and relapse definitions (S1 Appendix, S1 Text). The seasonality of primary infections was pronounced (Fig 1a and 1f). In contrast, the seasonality for relapses was less marked. The predicted peaks for the relapses appeared in different time intervals compared to primary infections (Fig 1).

To reflect uncertainty in the duration of blood-stage infection, we used two alternative input parameter value sets (S1 Appendix). Parameter set 1, which assumes a shorter mean duration of blood-stage infection, resulted in slightly higher estimates of incidence since a greater number of infections would be needed to achieve the same observed prevalence at the cross-sectional surveys.

In our cohort, the expected cumulative number of primary infections was roughly eight at one year of age to roughly 40 at four years old (S1 Fig).

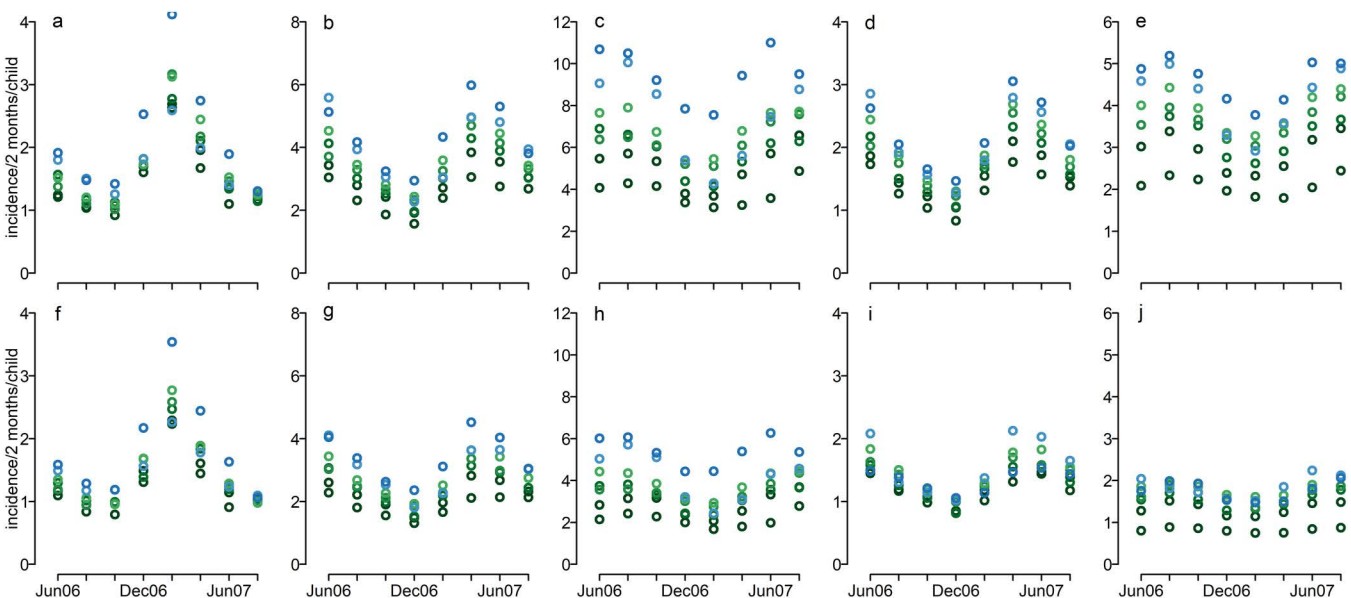

**Fig 1. The incidence of primary infection and relapse over calendar time and for each age group.** Top row: Input parameter value set 1 (shorter mean duration of blood-stage infection). Bottom row: Input parameter value set 2 (longer mean duration of blood-stage infection). Predicted incidence for children from Ilaita village with no ITN use of: a&f primary infections; b&g first or second relapse (relapse classification A2j); c&h third or later relapse (relapse classification A2j); d&i first relapse (relapse classification B2j); e&j second or later relapse (relapse classification B2j). Age categories (dark green through blue): 1-1.5 years, 1.5-2, 2-2.5, 2.5-3, 3-3.5, 3.5-4 light blue, over 4 years: blue.

## The estimated probability of clinical illness following primary infection and relapse

We validated the Bayesian statistical model using simulation. It was able to recover known parameter values for the probability of clinical illness, given correct inputs for the number of relapses, primary infections and the numbers of children and numbers of clinical cases (S1 Table).

We applied the statistical model to the Ilaita cohort clinical data and our expected numbers of primary infection and relapse. The predicted and observed clinical incidence values by age-group and by two-month interval were similar, suggesting reasonable model fit (Fig 2).

For all the model variants, the estimated probability of illness following *P. vivax* primary infection declined most rapidly in younger children (Fig 3). The model variants differed in the definitions of relapses assumed to be able to cause illness (Table 1) in order to reflect uncertainty. These differences lead to different estimated probabilities of illness. The probability of illness following relapse was lower if there were a higher number of relapses (relapse definition A included all relapses Fig 3A1 and 3A2) compared to counting only relapses occurring when no blood-stage infection by the same brood was present (relapse definition B) (Fig 3B1 and 3B2). For model variants which divided the relapses into those occurring earlier and later, the later relapses were estimated to have a lower probability of illness (Fig 3A2 and 3B2). The parameter estimates are in Table 3.

Varying seasonally, relapses were estimated to contribute roughly half of *P. vivax* illness in this cohort (Fig 4) despite accounting for 80% of the force of blood-stage infection. The overall estimated proportion of *P. vivax* illness contributed by relapses is consistent with that estimated in a trial of primaquine in another site in Papua New Guinea [19]. The seasonal variation in the proportions is also similar to that estimated from health facility data in Ethiopia [62], although this setting has different relapse patterns. The seasonality of *P. vivax* illness has been found to vary from that of *P. falciparum* [63], suggesting that relapses have a different seasonality to primary infections.

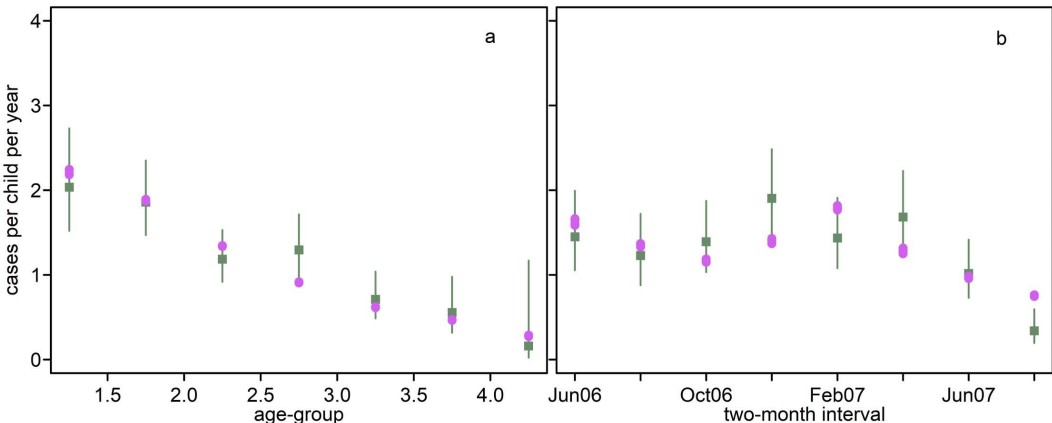

**Fig 2. Observed and predicted clinical incidence in the Ilaita cohort.** Green squares: observed clinical incidence with 95% CI; purple circles: predicted clinical incidence. Model predictions for four variants (A1, A2, B1, B2) are shown for two parameterizations for blood-stage infection duration each (purple circles are on top of each other). Each is a weighted sum of the estimated incidence for each covariate category multiplied by the number of children in that category. The patterns of observed and predicted incidence show good agreement, suggesting reasonable model fit.

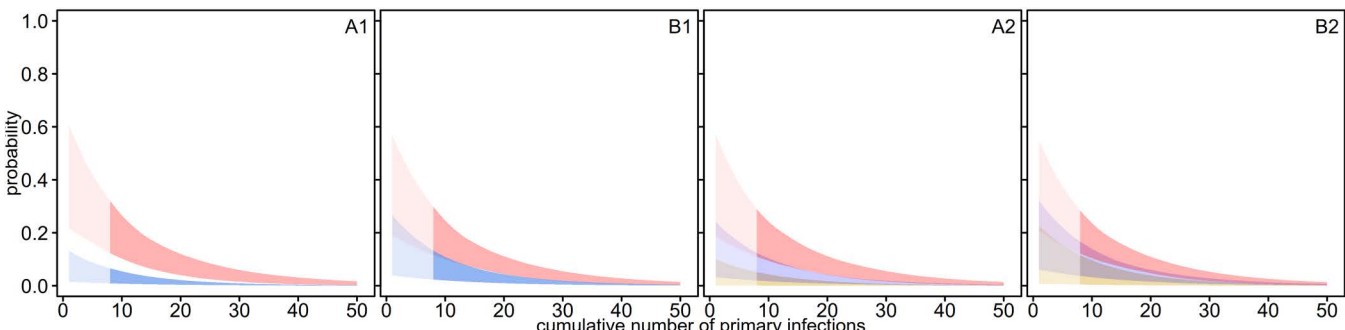

**Fig 3. The estimated probability of clinical illness following primary infections and relapses by cumulative number of primary infections.** Model variants A1, A2, B1, B2 are shown, all with all relevant relapses (j) (Model variants are described in Table 1). Relapse classification A (all relapses excluding those during a prophylactic period), Relapse classification B (excluding relapses occurring when there is a blood-stage infection by the same brood present or during an immune period after natural clearance or during a prophylactic period). Model variants A1 and B1: primary infections (red) and relapses as one group (blue); A2 and B2: primary infection (red), first relapse (lavender), second of later relapse (orange). Polygons are the minimum and maximum of the 95% credible intervals for the probability of clinical illness following blood-stage infection for the two parameter value sets (arising from two different assumptions about the duration of blood-stage infection - S1 Appendix). The youngest children in the cohort are already one year of age, so the darker shaded range where data are available starts only at seven cumulative primary infections. Maternal immunity is not included in this statistical model. Studies of deliberate infections studies estimate a high probability of clinical illness in naive individuals [132] but the probability is not known in natural conditions.

## Predicted incidence of clinical illness by age

To allow the prediction of age-incidence curves, we combined our model components for primary infection, relapse and clinical illness with components for the latent period prior to the primary infection, maternal immunity and mosquito biting. We validated the model variants using observational data from settings with the same relapse pattern (Fig 5). In cases where the key inputs (the transmission intensity and treatment coverage) were unknown, we used a range of values. In the Wosera, Papua New Guinea (Fig 5a), the transmission intensity was measured with 12 infectious bites per person per year in 1992 [64], and the observed age-incidence curve was estimated using data from health facilities in the years 1992–93 [22,44]. The proportion of the population who seek care when sick is not known

**Table 3. Estimated parameter values for the model for the probability of illness following primary infections and relapses.**

| | Relapse definition A | | Relapse definition B | |
|---|---|---|---|---|
| | With shorter blood-stage duration 1[6] | With longer blood-stage duration 2[7] | With shorter blood-stage duration[6] | With longer blood-stage duration[7] |
| | *Model variant A1*[1] | | *Model variant B1*[3] | |
| $p_{p0}$ | 0.38 (0.23, 0.56) | 0.44 (0.25, 0.67) | 0.36 (0.21, 0.54) | 0.41 (0.25, 0.62) |
| $p_{r0}$ | 0.05 (0.02, 0.08) | 0.08 (0.02, 0.15) | 0.10 (0.04, 0.17) | 0.18 (0.07, 0.30) |
| $X^*$ | 0.08 (0.06, 0.09) | 0.09 (0.07, 0.11) | 0.08 (0.06, 0.09) | 0.09 (0.07, 0.11) |
| Ratio of probability for relapse compared to primary infection[5] | 0.13 (0.03, 0.31) | 0.20 (0.04, 0.52) | 0.31 (0.09, 0.68) | 0.49 (0.13, 0.93) |
| | *Model variant A2*[2] | | *Model variant B2*[4] | |
| $p_{p0}$ | 0.35 (0.20, 0.52) | 0.40 (0.24, 0.60) | 0.35 (0.21, 0.53) | 0.41 (0.27, 0.60) |
| $p_{r_1 0}$ | 0.09 (0.03, 0.19) | 0.14 (0.05, 0.25) | 0.15 (0.05, 0.27) | 0.22 (0.10, 0.34) |
| $p_{r_2 0}$ | 0.03 (0.002, 0.07) | 0.05 (0.002, 0.12) | 0.07 (0.01, 0.14) | 0.13 (0.01, 0.25) |
| $X^*$ | 0.08 (0.06, 0.09) | 0.09 (0.07, 0.10) | 0.08 (0.06, 0.09) | 0.09 (0.07, 0.10) |
| Ratio of $p_{r_1 0}$ to $p_{p0}$[5] | 0.29 (0.06, 0.78) | 0.37 (0.09, 0.85) | 0.46 (0.11, 0.91) | 0.56 (0.18, 0.96) |
| Ratio of $p_{r_2 0}$ to $p_{r_1 0}$[5] | 0.43 (0.02, 0.96) | 0.42 (0.01, 0.96) | 0.55 (0.05, 0.98) | 0.61 (0.07, 0.98) |

Parameter estimates and 95% credible intervals.

[1] Model variant A1 includes all relapses occurring outside treatment prophylactic periods (with relapse definition A).

[2] Model variant A2 includes first and second; and third or later relapses (with relapse definition A).

[3] Model variant B1 includes relapses occurring in the absence of blood-stage infection by the same brood, immune period (20 days after natural clearance) and treatment prophylactic period (with relapse definition B).

[4] Model variant B2 categorizes relapses into first; and second or later (with relapse definition B).

[5] The ratios do not change by the number of cumulative primary infections since we estimate only one constant $X^*$.

[6] input parameter value set 1 includes the shorter mean duration of blood-stage infection.

[7] input parameter value set 2 includes the longer mean duration of blood-stage infection.

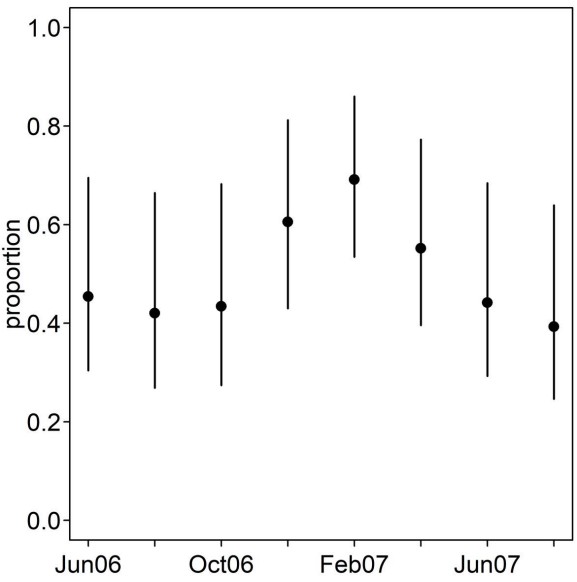

**Fig 4. The estimated proportion of *P. vivax* illness due to primary infections by season.** The estimated proportion (95% credible interval) for one model variant (B2j) and one parameter set (long duration parameter set 2) for one age-group (3-3.5 years).

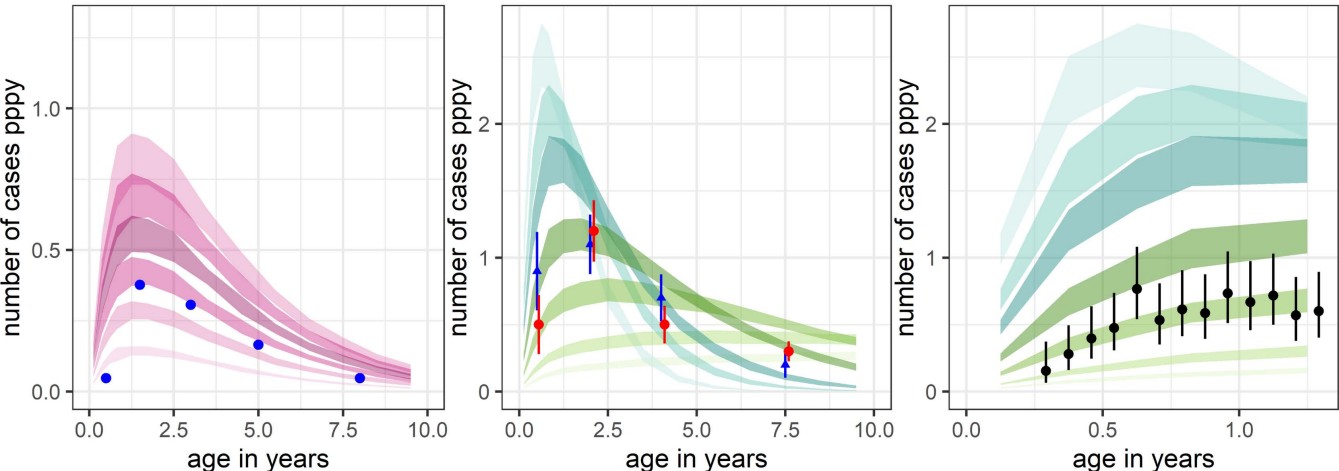

**Fig 5. Observed and predicted age-incidence curves from the same geographical zone with the same relapse pattern.** (a) Blue dots: Incidence of clinical cases presenting to two health facilities in residents of 29 villages in the Wosera, Papua New Guinea [22,44], an area with an estimated 12 infectious bites pppy in 1992 [24,55] Pink ribbons: minimum and maximum predicted incidence for the four model variants with 12 infectious bites pppy for treatment coverages (from lowest polygon to highest) of 60%,50%,40%,30%,20%,10% (actual treatment coverage is unknown). (b) Observed incidence in Espiritu Santu, Vanuatu in 1992-93 (blue triangles) and 1993-94 (red circles) with 95% confidence intervals [57]. Cases were detected through weekly active surveillance and visits were not included if the individual had had anti-malaria treatment in the last 4 weeks. Green ribbons: Predicted age-incidence curves using the minimum and maximum of the four model variants assuming all cases are detected for infectious bites pppy of 50,30,20,10,5,2,1 (highest peak polygon to lowest peak). (c) Incidence of *P. vivax* fevers in infants with 95% confidence intervals. The observed data are from the placebo arm of a trial of intermittent preventive treatment in infants (IPTi) trial, Mugil, Papua New Guinea [60]. Infants were followed up from 3 to 15 months of age. Passive case detection was carried out at one health centre and three outlying clinics. *P. vivax* illness is defined as a history of fever in the last 48 hours or axillary temperature ≥37.5 with a positive blood smear. For the present study, we do not exclude from the time at risk the period within 28 days of an illness but do exclude a prophylactic period of 14 days following treatment with artemether/lumefantrine (co-artem). Predicted age-incidence curves are the same as for (b), varying by transmission intensity. The actual transmission intensity and treatment coverage are not known, but treatment coverage is expected to be high. For all scenarios, we assume stable transmission with seasonal fluctuations as for Ilaita.

however and so we predicted age-incidence curves for different levels of treatment-seeking. There was reasonable agreement for observed data and predicted curves with care sought for 30% of *P. vivax* cases. The study in Vanuatu (Fig 5b) used weekly active case detection and so we assume that all cases were detected. However, the transmission intensity is not known and so we simulated a range of values. The agreement was good for 10 and 20 infectious bites per person per year. Infants in the placebo arm of a trial of IPTi were followed up using passive case detection (Fig 5c). Since neither the transmission intensity nor the proportion seeking care are known, the scope for validation is limited. Nonetheless both the observed and simulated age-incidence curves are consistent with the greatest impact of maternal immunity occurring in the early months.

Our predicted probabilities of illness decrease with increasing acquired clinical immunity. Taking cumulative exposure and detectability of infections into account, our predictions are consistent with estimates from studies in low transmission settings in the Amazon which indicated higher probabilities than in our cohort setting [26,27,65]. The low predicted incidence of *P. vivax* illness in older children (Fig 5b) is supported by observations from cohorts which include older ages in Papua New Guinea [19,58,59].

The different model variants gave similar predicted age-incidence curves (Fig 5): this was expected as they were fitted to the same data and due to their structure would only be expected to differ substantially with the introduction of some interventions triggered by illness from the primary infection.

## Discussion

We estimated the probability of illness following *P. vivax* primary infection and relapse in an endemic setting in Papua New Guinea. To our knowledge, these are the first estimates for both primary infections and relapses accounting for previous exposure. We found that the probability of illness for primary infections and relapses declines exponentially with the cumulative number of primary infections. The probabilities for illness following relapse were lower than for primary infection and depended on assumptions about whether all or only some relapses could cause illness. Later relapses of the same brood tended to have lower probabilities than earlier ones. Varying seasonally, relapses were estimated to contribute half of *P. vivax* illness in this cohort despite accounting for a previously estimated 80% of the force of blood-stage infection. Our results imply that interventions triggered by clinical illness would focus on more recent infections and age-groups with less acquired immunity, and that clearing hypnozoites when treating illness caused by primary infections would reduce clinical illness from relapses.

This study aims to inform the parameterization of an individual-based stochastic simulation model of *P. vivax* dynamics. The inclusion of clinical illness in a model will enable predictions of the likely impacts and costs of intervention strategies that are triggered by clinical episodes, such as case-management strategies and targeted interventions [66]. While many mathematical models of *P. vivax* dynamics do not include clinical illness since it is not relevant for their aim [19,67–95], there are several that do. Some models, particularly for low transmission settings, have assumed that every primary infection or every blood-stage infection is initially symptomatic [96–107]. Other models allow a constant probability of illness from blood-stage infection [108–115], or allow the probability to vary for a few [27,116–121] or many [27,30,69,122–128] levels of acquired clinical immunity. The acquired clinical immunity is gained through exposure or age, and a few models also allow this to decay. Some models allow the probability of illness to vary explicitly by primary infection and relapse [119,120]. A few models were fitted to data to estimate the clinical parameters [27,30,69,96,106–109,116,117,121]. In the present study, we estimate the probabilities of illness following both primary infection and relapses and by the number of previous primary infections as a proxy for acquired immunity.

There are several limitations in our estimates of both the numerator (the number of cases of illness) and the denominators (the numbers of primary illness and relapses). The definitions for *P. vivax* cases were adopted from previous studies [23,24] which estimated a pyrogenic threshold parasitaemia with >99% sensitivity and around 90% specificity for identifying *P. vivax* episodes in the study population [24]. Altering this cut-off or adjusting for the frequency of active case detection would alter the numbers of cases and consequently the estimated probabilities of illness. When estimating the numbers of primary infections and relapses, we aimed to make as few assumptions as possible and to base them on data from the same geographical area with the same relapse pattern however some assumptions were still required. The uncertainty in the force of primary infection was not included in the simulation, but it was estimated with reasonable precision (11.5 (95% CI 10.5, 12.3) for parameter set 2 or 13.3 (11.9, 14.2) for parameter set 1 [31]). The incidence of primary infection was estimated from genotyping data, although it is possible that relapses that appear as heterologous could be related as meiotic siblings derived from the same ookinete [39]. We acknowledge that some assumptions such as a perfectly repeating seasonal pattern are unlikely to be correct. Other assumptions based on data are the timing of relapses (from deliberately infected prison volunteers with the Chesson strain ([43] and S1 Appendix), and the timing of clinical illness (we assume that it occurs shortly after patent blood-stage infection [129]).

We assume independence between broods, and that relapses are independent of other hypnozoites or non-circulating merozoites within the same brood. There is some support for the assumption of independence of broods from a recent study in Indonesian soldiers returning from malarious areas: evidence of independent activation was found with a large proportion of relapses with only a single genotype in individuals with multiple parasite strains in their livers [39]. In addition, we do not take into account the possibility that the risk of illness could be different at the end of the transmission season compared to the start. Infections may weaken the host or boost immunity, or co-morbidity may be implicated [130]. The frequency of blood-stage infection in this setting at the time of the cohort was high and so we could not estimate a

decay of acquired clinical immunity. We also do not take into account incidents such as fever which may trigger relapses [131] or heterogeneity between children within their covariate category.

In future, further estimates from settings with different relapse patterns, transmission intensities and other characteristics would inform generalizability. The same study design, however, would be impractical in areas with low transmission and relatively few clinical cases.

## Conclusions

The estimated probabilities of illness following primary infection and relapse declined exponentially with the cumulative number of primary infections. These findings suggest that acquired clinical immunity is gained rapidly with early infections, that the probability of illness following relapse is 12% to 56% of that of primary infection depending on definition, and that later relapses have a lower probability of illness than earlier ones in the same brood. Our results can inform estimates of the burden of *P. vivax* and provide building blocks for mathematical models for predicting the impact of interventions. Interventions that target only clinical cases will focus on primary infections and early relapses and age groups with less acquired clinical immunity.

## Supporting information

**S1 Appendix. Simulation model.**
(DOCX)

**S1 Text. The incidence of antimalarial treatment in the cohort.**
(DOCX)

**S1 Fig. The expected cumulative number of primary infections by age-group and time interval.**
(DOCX)

**S1 Table. Ability of the statistical model to recover known parameter values for the probabilities of illness following primary infection and relapse from simulated data.**
(DOCX)

## Acknowledgments

We thank Olivier Briët, Aurélien Cavelan, Nakul Chitnis, Federica Giardina, Marek Kwiatkowski, Christian Schindler, Erin Stuckey, Thomas Smith, Rahel Wampfler and Lisa White for helpful comments. Calculations were performed at sciCORE (http://scicore.unibas.ch/) scientific computing center at the University of Basel.

## Author contributions

**Conceptualization:** Amanda Ross, Cristian Koepfli, Ivo Mueller.

**Data curation:** Cristian Koepfli, Lincoln Timinao, Benson Kiniboro.

**Formal analysis:** Amanda Ross.

**Funding acquisition:** Amanda Ross, Ingrid Felger, Ivo Mueller.

**Methodology:** Amanda Ross, Cristian Koepfli, Ingrid Felger.

**Software:** Amanda Ross, Diggory Hardy, Tobias Thüring.

**Validation:** Melanie Loeffel.

**Writing – original draft:** Amanda Ross.

**Writing – review & editing:** Amanda Ross, Cristian Koepfli, Lincoln Timinao, Diggory Hardy, Tobias Thüring, Melanie Loeffel, Benson Kiniboro, Ingrid Felger, Ivo Mueller.

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
