## [Editor Report · Decision Letter 0]

27 Feb 2024

Dear Dr. Ross,

Thank you very much for submitting your manuscript "The probability of Plasmodium vivax acute illness following primary infection and relapse in Papua New Guinea" (PNTD-D-24-00010) for consideration at PLOS Neglected Tropical Diseases. As with all papers peer reviewed by the journal, your manuscript was reviewed by members of the editorial board and by several independent peer reviewers. Based on the reports, we regret to inform you that we will not be pursuing this manuscript for publication at PLOS Neglected Tropical Diseases.

Based on the input from the reviewers this data does not add significantly to the existing knowledge regarding P. vivax infection and relapse.

The reviews are attached below this email, and we hope you will find them helpful if you decide to revise the manuscript for submission elsewhere. We are sorry that we cannot be more positive on this occasion. We very much appreciate your wish to present your work in one of PLOS's Open Access publications.

Thank you for your support, and we hope that you will consider PLOS Neglected Tropical Diseases for other submissions in the future.

Sincerely,

Hira L Nakhasi, Ph.D.

Section Editor

Hira Nakhasi

Section Editor

Shaden Kamhawi

co-Editor-in-Chief

Paul Brindley

co-Editor-in-Chief

---

## [Decision Letter · Decision Letter 1]

26 Aug 2024

Dear Dr. Ross,

Thank you very much for submitting your manuscript "The probability of Plasmodium vivax acute illness following primary infection and relapse in Papua New Guinea" for consideration at PLOS Neglected Tropical Diseases. As with all papers reviewed by the journal, your manuscript was reviewed by members of the editorial board and by several independent reviewers. In light of the reviews (below this email), we would like to invite the resubmission of a significantly-revised version that takes into account the reviewers' comments.

It seems to me important that the recommendations made by Reviewer 2 are fully implemented. The lack of clarity in the manner the mathematical modelling has been undertaken and a confusing set of figures will both be of great disservice to the ultimate message of the work.

With best regards

Georges

We cannot make any decision about publication until we have seen the revised manuscript and your response to the reviewers' comments. Your revised manuscript is also likely to be sent to reviewers for further evaluation.

Sincerely,

Georges Snounou, Ph.D.

Guest Editor

Hira Nakhasi

Section Editor

It seems to me important that the recommendations made by Reviewer 2 are fully implemented. The lack of clarity in the manner the mathematical modelling has been undertaken and a confusing set of figures will both be of great disservice to the ultimate message of the work.

With best regards

Georges

Reviewer's Responses to Questions

**Key Review Criteria Required for Acceptance?**

**Methods**

-Are the objectives of the study clearly articulated with a clear testable hypothesis stated?

-Is the study design appropriate to address the stated objectives?

-Is the population clearly described and appropriate for the hypothesis being tested?

-Is the sample size sufficient to ensure adequate power to address the hypothesis being tested?

-Were correct statistical analysis used to support conclusions?

-Are there concerns about ethical or regulatory requirements being met?

Reviewer #1: The author's methods are clear and can easily be repeated by anyone with knowledge in the field. The authors did a fantastic job accounting for the large variability associated with predictions of this nature by including several parameters and conditions for the simulations. The sample size is sufficient for a baseline prediction model and the statistics used are sound.

Reviewer #2: This paper tackles the important problem of assessing how relapses contribute to clinical episodes of P. vivax malaria. It has been hypothesized that relapses are less likely to cause clinical malaria than primary infections. Although this hypothesis is consistent with a number of experimental observations, and makes “biological sense”, it has not been definitively demonstrated. Furthermore, there are important implications for mathematical models of P. vivax transmission, as most published models to do not account for potential differences in the probability of illness between primary infections and relapses.

The underlying epidemiological data and genotyping data are solid. The modelling approach is appropriate, and the presented results are all credible.

Although I do not doubt any of the findings, my main concern relates to the presentation of the methods and results. Put simply, they’re very hard to follow.

For the benefit of the editorial team, I will lay out my understanding of the methodology (the authors should feel free to clarify any misunderstandings I may make). The starting data is from a cohort study in Ilaita in PNG. Blood samples from this study were genotyped using a single locus genotyping marker. These genotyping data were modelled to provide estimates of seasonally varying force of blood-stage infection over time, stratified by primary infection and relapse. This current paper builds on all of this past work and takes the previous model estimates of incidence as input. Furthermore, two different model estimates corresponding to different assumed duration of blood-stage infections are included.

Table 1 provides a description of 8 different models according to how relapses are defined, and assumptions regarding the probability that a relapse causes clinical illness.

The model also accounts for seasonality, age and covariates such as ITN use. A proxy of lifetime immunity is calculated by extrapolating backwards to estimate the total number of lifetime infections.

The OpenMalaria model is utilised to incorporate different model components and provide predictions of age-incidence curves. I really struggled to understand how the OpenMalaria model was used. When I follow the citation [39], it just says OpenMalaria and no other information is provided. OpenMalaria is a very impressive and well documented framework for mathematical modelling of malaria transmission, but to my knowledge there has been no published work on the use of P. vivax. Therefore, it’s not clear how it’s being used in this instance.

In summary, I have a very positive opinion of this important work, I am in agreement with the findings, but I have some substantial reservations about the presentation of the methods. A recommendation to the authors would be to simplify the presentation of the results by selecting one model variant and one set of model assumptions for the main manuscripts, and retain the others for the supplementary information.

The figures are also very hard to follow. They have multiple panels, different scales, no labels, and often no legends. The work as a whole would be easier to follow if the figures were simplified and made more consistent.

Reviewer #3: The methods of this study are sound and bolstered by a strong evidence base. The data from the cohort study and region is general is very rich.

A minor question in SI section (iii) treatment - is the treatment time window sampled stochastically 2 weeks?

Main text classifying relapse - specify how early and late relapse are defined?

Clarify the sentence that begins, "if there was selfing." Does the greatest genetic variation there refer to in the brood?

**Results**

-Does the analysis presented match the analysis plan?

-Are the results clearly and completely presented?

-Are the figures (Tables, Images) of sufficient quality for clarity?

Reviewer #1: All results are clearly labelled and illustrate the data in a logical way.

Reviewer #2: (No Response)

Reviewer #3: A minor suggestion to label plots (particularly Fig 1).

Fig 4 - in addition to this or a call out in the text about the proportion of symptomatic Pv illness due to relapse would also be useful for policy/advocacy purposes.

**Conclusions**

-Are the conclusions supported by the data presented?

-Are the limitations of analysis clearly described?

-Do the authors discuss how these data can be helpful to advance our understanding of the topic under study?

-Is public health relevance addressed?

Reviewer #1: The authors do a fantastic job at relating the findings to public health and are upfront with the limitations of their study. This is a great estimation in P. vivax relapse.

Reviewer #2: (No Response)

Reviewer #3: Yes- the conclusions are clear and it is very useful to have the findings of a modeling study anchored in the policy implications (ie, what groups to prioritize for treatment)

**Editorial and Data Presentation Modifications?**

Reviewer #1: There are a few typos present in your manuscript, please double check the grammar prior to proof approval.

Reviewer #2: (No Response)

Reviewer #3: I have provided minor modification suggestions above.

**Summary and General Comments**

Reviewer #1: (No Response)

Reviewer #2: (No Response)

Reviewer #3: The authors address an important topic using a complex but robust analytical framework and anchor it to the implications of policy.

PLOS authors have the option to publish the peer review history of their article (what does this mean? ). If published, this will include your full peer review and any attached files.

**Do you want your identity to be public for this peer review?** For information about this choice, including consent withdrawal, please see our Privacy Policy .

Reviewer #1: No

Reviewer #2: No

Reviewer #3: No
---

## [Decision Letter · Decision Letter 2]

27 Mar 2025

PNTD-D-24-00010R2The probability of Plasmodium vivax acute illness following primary infection and relapse in Papua New GuineaPLOS Neglected Tropical Diseases  Dear Dr. Ross, Thank you for submitting your manuscript to PLOS Neglected Tropical Diseases. After careful consideration, we feel that it has merit but does not fully meet PLOS Neglected Tropical Diseases's publication criteria as it currently stands. Therefore, we invite you to submit a revised version of the manuscript that addresses the points raised during the review process. Please submit your revised manuscript within 30 days Apr 26 2025 11:59PM. If you will need more time than this to complete your revisions, please reply to this message or contact the journal office at plosntds@plos.org.  Please include the following items when submitting your revised manuscript: * A rebuttal letter that responds to each point raised by the editor and reviewer(s). You should upload this letter as a separate file labeled 'Response to Reviewers '. This file does not need to include responses to any formatting updates and technical items listed in the 'Journal Requirements' section below. * A marked-up copy of your manuscript that highlights changes made to the original version. You should upload this as a separate file labeled 'Revised Manuscript with Track Changes '. * An unmarked version of your revised paper without tracked changes. You should upload this as a separate file labeled 'Manuscript '. If you would like to make changes to your financial disclosure, competing interests statement, or data availability statement, please make these updates within the submission form at the time of resubmission. Guidelines for resubmitting your figure files are available below the reviewer comments at the end of this letter. We look forward to receiving your revised manuscript. Kind regards,

Gregory Deye

Academic Editor

Susan Madison-AntenucciSection EditorPLOS Neglected Tropical Diseases

Shaden Kamhawi

co-Editor-in-Chief

Paul Brindley

co-Editor-in-Chief

**Reviewers' comments:** 

Reviewer's Responses to Questions

**Key Review Criteria Required for Acceptance?**

**Methods:**

-Are the objectives of the study clearly articulated with a clear testable hypothesis stated?

-Is the study design appropriate to address the stated objectives?

-Is the population clearly described and appropriate for the hypothesis being tested?

-Is the sample size sufficient to ensure adequate power to address the hypothesis being tested?

-Were correct statistical analysis used to support conclusions?

-Are there concerns about ethical or regulatory requirements being met?

Reviewer #2: I will keep my comments here within the scope of my comments from my previous review. I previously wrote that I had a positive assessment of the importance of this work, and I maintain this opinion. My primary concern was that the models as presented in the manuscript were very difficult to understand, and I maintain this opinion.

In response to a recommendation that the authors clarify the description of the model methods, the authors have made some small edits on page 5.

In response to the recommendation to “simplify the presentation of the results by selecting one model variant and one set of model assumptions for the main manuscripts, and retain the others for the supplementary information” the authors have chosen to retain all 8 model variants. This is their choice and I won’t argue further against it.

In response to the comment about the figures being difficult to follow, the authors have made no change other than to copy and paste the existing figure legends to pages 19 and 20 of the revised manuscript. In the absence of any change, I maintain that the figures are still very difficult to understand.

In response to my comment on the difficulty of understanding the interaction with OpenMalaria, the authors have responded that more information is available at a github repository. However, reference 39 still just reads ‘OpenMalaria’, and the github repository is not cited anywhere in the manuscript – only in the response to reviewers. This is certainly not a helpful response.

**Results:**

-Does the analysis presented match the analysis plan?

-Are the results clearly and completely presented?

-Are the figures (Tables, Images) of sufficient quality for clarity?

Reviewer #2: (No Response)

**Conclusions:**

-Are the conclusions supported by the data presented?

-Are the limitations of analysis clearly described?

-Do the authors discuss how these data can be helpful to advance our understanding of the topic under study?

-Is public health relevance addressed?

Reviewer #2: (No Response)

**Editorial and Data Presentation Modifications?**

Reviewer #2: (No Response)

**Summary and General Comments:**

Reviewer #2: (No Response)

PLOS authors have the option to publish the peer review history of their article (what does this mean? ). If published, this will include your full peer review and any attached files.

**Do you want your identity to be public for this peer review?** For information about this choice, including consent withdrawal, please see our Privacy Policy .

Reviewer #2: No

---

## [Editor Report · Decision Letter 3]

15 Sep 2025

Dear Dr. Ross,

We are pleased to inform you that your manuscript 'The probability of Plasmodium vivax acute illness following primary infection and relapse in Papua New Guinea' has been provisionally accepted for publication in PLOS Neglected Tropical Diseases.

We appreciate the thorough response to the comments of the reviewer. We are accepting this manuscript for publication (congratulations), but there are couple of changes  of a typographical nature that should be addressed before publication.

Comments (line numbers refer to the R3 submission PDF):

1) Ln 47- 600,000 is the number for total malaria deaths in 2023. The number for P. vivax is much smaller. I recognize that this is likely an oversight caused by reconstruction of a sentence that might have originally referred to all malaria. (this is actually the primary reason for this email as I'm sure the authors would like to correct this. Subsequent comments are really only since they will be submitting a change anyway).

2) Ln 68-"Removed" is strange and usually refers to a physical process. The references used PQ treatment. Would prefer "...treated some participants to eliminate hypnozoites reported..."

3) Ln 174- "selfing"-Might not be generally known by readers of epidemiologic papers. It would be nice give a brief definition parenthetically.

Best regards,

Gregory Deye

Academic Editor

Susan Madison-Antenucci

Section Editor

Shaden Kamhawi

co-Editor-in-Chief

Paul Brindley

co-Editor-in-Chief

I appreciate the thorough response to the comments of the reviewer. My intent is to accept this manuscript for publication (congratulations), but there are couple of changes of a typographical nature that I would suggest/request before publication. I apologize about this awkward method of communicating these requests, but this seems to be the only method that the Editorialmanager software will allow. I would have preferred a direct email communication.

Comments (line numbers refer to the R3 submission PDF):

1) Ln 47- 600,000 is the number for total malaria deaths in 2023. The number for P. vivax is much smaller. I recognize that this is likely an oversight caused by reconstruction of a sentence that might have originally referred to all malaria. (this is actually the primary reason for this email as I'm sure the authors would like to correct this. Subsequent comments are really only since they will be submitting a change anyway).

2) Ln 68-"Removed" is strange and usually refers to a physical process. The references used PQ treatment. Would prefer "...treated some participants to eliminate hypnozoites reported..."

3) Ln 174- "selfing"-Might not be generally known by readers of epidemiologic papers. It would be nice give a brief definition parenthetically.

I look forward to your response.

---

## [Editor Report · Acceptance letter]

Dear Dr. Ross,

We are delighted to inform you that your manuscript, " 

The probability of Plasmodium vivax acute illness following primary infection and relapse in Papua New Guinea," has been formally accepted for publication in PLOS Neglected Tropical Diseases.

Best regards,

Shaden Kamhawi

co-Editor-in-Chief

Paul Brindley

co-Editor-in-Chief
